DZNep, an inhibitor of the histone methyltransferase EZH2, suppresses hepatic fibrosis through regulating miR-199a-5p/SOCS7 pathway

Ding Rongrong 1 2
Zheng Jianming 1
Li Ning 1
Cheng Qi 1
Zhu Mengqi 1
Wang Yanbing 2
Zhou Xinlan 2
Zhang Zhanqing doctorzzqsphc@163.com 2
Shi Guangfeng gfshi2005@163.com 1
1 Infectious Diseases, Huashan Hospital, Fudan University , Shanghai , China
2 Hepatobiliary Medicine, Shanghai Public Health Clinical Center, Fudan University , Shanghai , China
Lavia Patrizia
Electronic publication date: 2021 May 14
Publication date: 2021
Volume: 9
Electronic Location ID: e11374
Received 2020 Oct 12; Accepted 2021 Apr 7
Copyright: ©2021 Ding et al.
Copyright year: 2021
Copyright holder: Ding et al.
License: This is an open access article distributed under the terms of the Creative Commons Attribution License, which permits unrestricted use, distribution, reproduction and adaptation in any medium and for any purpose provided that it is properly attributed. For attribution, the original author(s), title, publication source (PeerJ) and either DOI or URL of the article must be cited.
License URL: https://creativecommons.org/licenses/by/4.0/

Keywords: Hepatic fibrosis, DZNep, TGF-β1, EZH2, miR-199a-5p, SOCS7

Funding: National Natural Science Foundation of China 81371821 Natural Science Foundation Project of Shanghai 19ZR1407800 This work was mainly supported by the National Natural Science Foundation of China (award number 81371821) and the Natural Science Foundation Project of Shanghai (NO. 19ZR1407800). The funders had no role in study design, data collection and analysis, decision to publish, or preparation of the manuscript.

==============================
Background

Hepatic fibrosis is a common response to chronic liver injury. Recently, the role of DZNep (a histone methyltransferase EZH2 inhibitor) in repressing pulmonary and renal fibrosis was verified. However, the potential effect of DZNep on hepatic fibrosis has not been elucidated.

Methods

The hepatic fibrosis model was established in rats treated with CCl4 and in hepatic stellate cells (HSCs) treated with TGF-β1. The liver tissues were stained with H&E and Masson’s trichrome. The expression of EZH2, SOCS7, collagen I, αSMA mRNA and miR-199-5p was assessed using qPCR, immunohistochemical or western blot analysis. A dual-luciferase reporter assay was carried out to validate the regulatory relationship of miR-199a-5p with SOCS7.

Results

The EZH2 level was increased in CCl4-treated rats and in TGF-β1-treated HSCs, whereas DZNep treatment significantly inhibited EZH2 expression. DZNep repressed hepatic fibrosis in vivo and in vitro, as evidenced by the decrease of hepatic fibrosis markers (α-SMA and Collagen I). Moreover, miR-199a-5p expression was repressed by DZNep in TGF-β1-activated HSCs. Notably, downregulation of miR-199a-5p decreased TGF-β1-induced expression of fibrosis markers. SOCS7 was identified as a direct target of miR-199a-5p. The expression of SOCS7 was decreased in TGF-β1-activated HSCs, but DZNep treatment restore d SOCS7 expression. More importantly, SOCS7 knockdown decreased the effect of DZNep on collagen I and α SMA expression in TGF-β1-activated HSCs.

Conclusions

DZNep suppresses hepatic fibrosis through regulating miR-199a-5p/SOCS7 axis, suggesting that DZNep may represent a novel treatment for fibrosis.

Introduction

Hepatic fibrosis, induced by pro-inflammatory cytokines, chemokines, or viral infection, causes about one million deaths annually worldwide due to the development of cirrhosis (Ezhilarasan, Sokal & Najimi, 2018; Lee, Wallace & Friedman, 2015). However, there is currently no effective anti-fibrotic therapy (Bollong et al., 2017). The hepatic stellate cells (HSCs) are the major source of extracellular matrix (ECM) proteins in liver fibrosis, including fibronectin (FN), α-smooth muscle actin (α-SMA), and Type I collagen (COL I) (Ellis & Mann, 2012; Pellicoro et al., 2014; Tacke & Trautwein, 2015). Despite commonly progressing to hepatic cirrhosis and organ damage, in principle, hepatic fibrosis is a reversible process. Emerging studies showed that liver fibrosis may be reversed or slowed when hepatic damage is successfully treated (Latasa et al., 2010; Tangkijvanich & Yee Jr, 2002).

The role of the enhancer of zeste homolog 2 (EZH2) in regulating myofibroblast transformation and tissue fibrosis has recently been verified (Xiao et al., 2016). Tsou et al. (2019) demonstrated that EZH2 expression is increased in scleroderma dermal fibroblasts and observed that forced expression of EZH2 in fibroblasts results in a pro-fibrotic response, whereas pharmacological inhibition of EZH2 effectively represses scleroderma fibrosis (Tsou et al., 2019). The EZH2 expression is upregulated in atrial fibroblasts, and pharmacological or genetic inhibition of EZH2 represses atrial fibroblast differentiation and results in a decrease in ECM production (Song et al., 2019). The role of EZH2 in regulating liver fibrosis is gradually being elucidated. The EZH2 expression is also elevated in a murine model of liver fibrosis and in TGF-β1-activated HSCs (Martin-Mateos et al., 2019). Inhibition of EZH2 decreases α-SMA expression in TGF- β1-activated HSCs by de-repressing DKK-1, an antagonist of Wnt signaling (Yang et al., 2017). Atta et al. (2014) showed that the EZH2 expression level is increased in fibrotic liver tissues, but HGF (hepatocyte growth factor) treatment reduces EZH2 expression and represses hepatic fibrosis in rats. Furthermore, an EZH2 inhibitor, DZNep, has been shown to play an important role in repressing liver fibrosis. Zeybel et al. demonstrated that DZNep presents an anti-fibrotic effect in a murine model of liver fibrosis and in TGF-β1-activated HSCs (Zeybel et al., 2017).

MicroRNAs (miRNAs) are a class of small single-stranded and evolutionary conserved noncoding RNA molecules (Ambros, 2001) (Eulalio, Huntzinger & Izaurralde, 2008; Liu, Fortin & Mourelatos, 2008). Emerging evidence suggests that miRNAs regulate almost all physiological and pathological processes, such as cell differentiation (Ortega et al., 2010), embryonic development (Laurent, 2008), tumorigenesis (Kumar et al., 2007), and tissue fibrosis (Patel & Noureddine, 2012). Several functional miRNAs in liver fibrosis have been identified, including miR-29, miR-21, miR-7-5p, miR-33a, and miR-129-5p (O’Reilly, 2016). Suppressor of cytokine signaling (SOCS) protein family (contains SOCS1-SOCS7) functions as a negative regulator of cytokine receptor signaling (Duncan et al., 2017). SOCSs play an important role in various physiological processes including inflammation, cell viability, and development (Durham et al., 2019; Fu et al., 2020). Emerging studies demonstrated the role of SOCS7 in regulating cancer cells proliferation (Noguchi et al., 2013), or in regulating insulin action (Banks et al., 2005). However, the function of SOCS7 on regulating inflammation and fibrosis remains unclear.

Based on above findings, here we investigated the role of DZNep in regulating hepatic fibrosis and uncovered the mechanism underlying DZNep slowing hepatic fibrosis. We demonstrated that the expression of miR-199a-5p was repressed by DZNep in TGF-β1-activated HSCs. Functionally, miR-199a-5p inhibition decreased TGF-β1-induced hepatic fibrosis, and knockdown of SOCS7 decreased the effect of DZNep on collagen I and α-SMA expression in TGF-β1-activated HSCs. These data showed that DZNep contributes to alleviate liver fibrosis through regulating miR-199a-5p/SOCS7 axis.

Materials and Methods

Animals and CCl4 and DZNep treatment

All experiments in this study were approved by the Animal Ethics Committee of Huashan Hospital, Fudan University (2018 Huashan Hospital JS-030). Male Sprague-Dawley (SD) rats (body weights: 180-200 g) were procured from Trophic Animal Feed High-Tech Co., Ltd. (Jiangsu, China) and kept under the specific-pathogen-free (SPF). CCl4 (20070721) and olive oil (060312) were obtained from Sinopharm Co., Ltd. (Shanghai, China). All rats were randomly divided into 3 groups: the control group (n = 5), CCl4-induced group (n = 5) and CCl4-induced+DZNep group (n = 5). The rats in the CCl4-induced group and CCl4-induced+DZNep group were injected intraperitoneally (i.p.) with 50% CCl4 olive oil solution at one mL/kg body weight two times a week (every Monday and Thursday) for 9 weeks, whereas the rats in the control group were subjected to conventional breeding. Two weeks later, in addition to CCl4, the CCl4-induced+DZNep group received DZNep (2 mg/kg, twice per week) by i.p. injection for an additional 6 weeks. Rats were sacrificed with CO2 narcosis, and their livers were subsequently removed, embedded in paraffin, and stained with H&E and Masson’s trichrome. Animals were kept in a colony room at controlled temperature (22 °C), and a 12:12 h light–dark cycle, with food and water available. All rats were sacrificed through overetherization and decapitation. 

Cell culture and TGF-β1 treatment

The human HSC line LX2 (ATCC, Manassas, VA, USA) was cultured in DMEM supplemented with 2% FBS (HyClone, Logan, UT, USA) at 37 ° C in a humidified atmosphere in 5% CO2. The cells were treated with 10 µg/L TGF-β1 (Peprotech, USA) in DMEM supplemented with 1% FBS for 48 h. DZNep (1 µmol/L) was added to treat cells for 24 h.

Cell transfection

MiR-199a-5p mimic, miRNA controls (miRcont), and miR-199a-5p inhibitors were obtained from RIBOBIO (Guangzhou, China). siRNA against SOCS7 (siSOCS7) and negative control (siNC) were obtained from Thermo Fisher (Waltham, MA, USA). All small RNAs (40nM) were transfected into LX2 cells, TGF-β1-activated LX2 cells in the presence or absence of DZNep with Lipofectamine™ 2000 (Invitrogen, Carlsbad, CA, USA) following manufacturer’s instructions, and then cells were collected to carry out qPCR or western blot analysis after 48 h.

Quantitative real-time PCR (qPCR)

For quantification of miR-199a-5p miRNA expression, cDNA was synthesized with a Reverse Transcription Kit (Thermo Fisher) and quantified with a miRNA RT-PCR Quantitation Kit (Qiagen, Duesseldorf, Germany). For quantification of EZH2, collagen I, α-SMA and SOCS7 mRNA expression, reverse transcription was performed with a High Capacity cDNA Reverse Transcription Kit (Thermo Fisher), and EZH2, collagen I, α-SMA and SOCS7 mRNA levels were measured with SYBR Green Mix (Vazyme, Nanjing, China). All primers employed were the following: miR-199a-5p: Forward: 5′-GCCAAGCCCAGTGTTCAGAC-3′, Reverse: 5′-GTGCAGGGTCCGAGGTATTC-3′. EZH2, Forward, 5′-CCCTGACCTCTGTCTTACTTGTGGA-3′; Reverse, 5′-ACGTCAGATGGTGCCAGCAATA-3′; Collagen I, Forward: 5′-CCTCAGGGTATTGCTGGACAAC-3′, Reverse, 5′-CAGAAGGACCTTGTTTGCCAGG-3′; α-SMA, Forward: 5′- GCGTGGCTATTCCTTCGTTA-3′, Reverse, 5′-ATGAAGGATGGCTGGAACAG-3′; SOCS7, Forward: 5′-ACAGGAAGGTTGGGATTCTC-3′, Reverse, 5′-CAGCACAGACTCTAACTCTG-3′.

Western blot analysis

Equal amounts of total protein (approximate 80  µg) were separated by a 12% SDS-polyacrylamide gel and transferred onto 0.2-µm PVDF membranes (Roche, Basel, Switzerland). The blots were incubated overnight at 4 ° C with antibodies (anti-EZH2: 1:5000, ab186006, Abcam, CA, USA; anti-Collagen I: 1:1000, ab6308, Abcam; anti- α-SMA: 1:1000, ab19245, CST; anti-SOCS7: 1:1000, ab224589, Abcam; anti- β-actin: 1:2000, ab8227, Abcam) and were subsequently incubated with a secondary antibody (1:5000, ab205718, Abcam). The protein signals were detected with ECL (Pierce Biotechnology, Rockford, IL) and digitized and analyzed densitometrically using ImageJ software.

Luciferase reporter assay

The recombinant plasmid of pGL3-SOCS7-3′-UTR (wildtype) or pGL3-SOCS7-3′-UTR-Mutation was constructed by inserting the 3′ UTR of SOCS7 or its mutant into the pGL3-promoter vector (Promega). LX2 cells (1 ×105 cells/well) were seeded in a 24-well plate and cotransfected with miR-199a-5p (40 nM) or miRcont, pGL3-SOCS7-3′-UTR (10 ng), and pRL-TK (1 ng, Promega) using Lipofectamine™ 2000. After 72 h, luciferase activity was assessed by Dual-Luciferase Assays Kit (Promega) according to the manufacturer’s protocol.

Immunohistochemistry

Tissues were fixed in 4% paraformaldehyde (PFA) in 4 ° C environments for 24 h. After that step, all tissues were dehydrated in the embedding box and embedded using liquid paraffin according to the manufacturer’s instructions (Sigma-Aldrich). The sections (measuring 4 µm) were prepared through a slicer and baked on slide warmers for 30 min. The sections were placed in a wet box, in which a small amount of distilled water was added in combination with 3% hydrogen peroxide and was incubated at room temperature for 10 min. After washing with PBS and distilled water 3 times each and for 3 min each time, goat serum occlusive fluid was added to the wet box to incubate sections for 10 min. Immunohistochemistry was performed by primary antibodies and the corresponding secondary antibody according to instruction procedures. Then, 100 µl DAB solution (Sigma-Aldrich) was added to each section and restained with hematoxylin for 1–2 min.

Immunofluorescence

LX2 cells were fixed with 4% paraformaldehyde for 15 min, followed by washing 3 times with sterile PBS at room temperature (RT). After blocking with 5% BSA (bovine serum albbumin) in 0.2% Triton X-100 for 1 h at RT, slides was incubated with anti-EZH2 primary antibody (1:200; ab191080; Abcam) overnight at 4 ° C, followed by incubating with goat anti-rabbit antibody (1:2000; A27039; Invitrogen). After washing 3 times, slides were mounted with glycerin and analyzed using a confocal microscopy (Zeiss LSM710, Jena, Germany). DAPI (Solarbio) was used to visualize the nucleus.

Statistical analysis

Data are presented as means  ± standard deviation (SD) from at least three independent experiments. The statistical analyses of differences between the two groups were carried out using Student’s t-test or one-way analysis of variance (ANOVA) followed by the Scheffé test after evaluating normality of data distribution. SPSS 17.0 (SPSS, Inc., USA) was applied to perform the statistical analysis. P < 0.05 was considered to indicate significance.

Results

DZNep inhibited CCl4-induced liver fibrosis in rats

To investigate the effects of DZNep on hepatic fibrosis, rats were treated with CCl4 in the presence or absence of DZNep. As shown in Figs. 1A and 1B, the results of H&E and Masson staining revealed that the liver tissues of the CCl4-treated group showed extensive necrosis, normal structure destruction and increased collagen deposition. However, DZNep treatment effectively ameliorated CCl4-induced pathologic lesions and collagen deposition. Furthermore, an immunofluorescence analysis was performed to assess the EZH2 protein level and its subcellular localization in TGF-β1-activated LX2 cells, and the data showed that EZH2 levels were significantly enhanced in TGF-β1-treated LX2 cells compared with control, whereas DZNep treatment inhibited the increase (Fig. 1C). Immunofluorescence analysis also showed that EZH2 was located mainly in the nucleus of HSCs (Fig. 1C).

Figure 1 DZNep resisted CCl_4induced liver fibrosis in rats.

The rat model was divided into three groups: (1) Control group; (2) CCl4-induced group; and (3) CCl4-induced+DZNep group. (A–B) The hepatic pathologic lesions and collagen deposition were assessed using H&E (A) and Masson (B) staining in the liver tissues of each group of rats (n = 5). (C) LX2 cells were treated with TGF-β1 (10 µg/L) in the presene or absence of DZNep (1 µmol/L), and then the expression and subcellular localization of EZH2 in LX2 cells was assessed using immunofluorescence analysis. Scan bar, 20 µm.

DZNep suppressed TGF-β1-induced LX2 cell activation

The LX2 cells were treated with TGF-β1 in the presence or absence of DZNep, and the antifibrosis effect of DZNep was assessed. Figure 2A showed that the EZH2 mRNA level was markedly upregulated in TGF-β1-activated LX2 cells compared with the control, but DZNep treatment notably repressed the enhancement. Then, collagen I and α-SMA expression were evaluated by qPCR and immunoblotting. The qPCR analysis data indicated that collagen I and α-SMA mRNA expression was significantly upregulated in TGF-β1-activated LX2 cells, whereas DZNep treatment repressed TGF-β1-induced upregulation of collagen I and α-SMA (Figs. 2B and 2C). Additionally, immunoblotting was performed to determine protein expression. The protein levels of EZH2, collagen I and α-SMA were also markedly upregulated in TGF-β1-activated LX2 cells, whereas DZNep treatment inhibited TGF-β1-induced upregulation of collagen I and α-SMA (Figs. 2D–2G). These data demonstrated that DZNep contributed to inhibit the expression of EZH2 and fibrosis markers in vivo and in vitro.

Figure 2 DZNep suppressed TGF-β1-induced LX2 cell fibrosis.

The LX2 cells were treated with TGF-β1 (10 µg/L) for 48 h in the presence or absence of DZNep (1 µmol/L) for 24 h. The cell model was divided into three groups: (1) Control group; (2) TGF-β1-activated group; and (3) TGF-β1-activated+DZNep group. (A–C) qPCR analysis of EZH2 (A), collagen I (B) and α-SMA (C) mRNA expression in each group of cells. (D–G) Representative western blot of EZH2 (D and E), collagen I (D and F), α-SMA (D and G) protein expression in each group of cells, and quantification of average across three separate experiments. ∗P < 0.05, ∗∗P < 0.01.

The effect of miR-199a-5p on fibrosis markers expression in vitro

We next investigated the mechanism governing the role of DZNep in fibrosis markers expression. Emerging research has demonstrated that EZH2 epigenetically regulates miRNA expression and that aberrant miRNAs are closely correlated with hepatic fibrosis (Feng et al., 2017; Huang et al., 2019; Kwon et al., 2017). To investigate whether DZNep alleviates fibrosis markers expression by regulating the expression of several miRNA, 13 hepatic fibrosis-related miRNAs (miR-29b, miR-34a, miR-34b, miR-15b, miR-16, miR-195, miR-15a, miR-199a, miR-200a, miR-200c, miR-378a, miR-424, and miR-29a) (Jiang et al., 2017) were analyzed using qPCR in TGF-β1-activated LX2 cells after DZNep treatment. Figure 3A showed that miR-199a-5p was the most significantly downregulated miRNA in TGF-β1-activated LX2 cells after DZNep treatment. Figure 3B further showed that the miR-199a-5p level was notably upregulated in LX2 cells after TGF-β1 treatment, whereas DZNep treatment inhibited this increase. DZNep also repressed CCl4-induced increase of miR-199a-5p in liver tissues (Supporting Fig. S1A). Next, we investigated the role of miR-199a-5p inhibition in fibrosis markers expression. The results from qPCR analysis showed that the miR-199a-5p inhibitor remarkably downregulated miR-199a-5p expression in TGF-β1-activated LX2 cells (Fig. 3C). Functionally, miR-199a-5p inhibition repressed TGF-β1-induced upregulation of collagen I and α-SMA mRNA (Fig. 3D and 3E). Western blot analysis further demonstrated that miR-199a-5p inhibition decreased TGF- β1-induced protein expression of collagen I and α-SMA in LX2 cells (Figs. 3F–3H).

Figure 3 The effect of miR-199a-5p on hepatic fibrosis in vitro.

(A) The expression level of 13 hepatic fibrosis-related miRNAs (miR-29b, miR-34a, miR-34b, miR-15b, miR-16, miR-195, miR-15a, miR-199a, miR-200a, miR-200c, miR-378a, miR-424 and miR-29a) was assessed using qPCR analysis in TGF-1-activated LX2 cells in the presence or absence of DZNep treatment (n = 3). (B) The expression level of miR-199a-5p was assessed using qPCR analysis in TGF-1-activated LX2 cells in the presence or absence of DZNep treatment. (C–E) The expression levels of miR-199a-5p (C), collagen I (D) and αSMA (E) were assesed using qPCR analysis in LX2 control cells compared with TGF-β1-activated LX2 cells treated with miRcont inhibitor or miR-199a-5p inhibitor. (F-H) Representative western blot of collagen I (F and G) and αSMA (F and H) protein expression in TGF-β1-activated LX2 cells treated with NC inhibitor or miR-199a-5p inhibitor, and quantification of average across three separate experiments. ∗P < 0.05, ∗∗P < 0.01.

SOCS7 was a direct target of miR-199a-5p

To illustrate the role of miR-199a-5p, the potential target genes of miR-199a-5p were screened by prediction websites (PicTar and TargetScan). SOCS7 was identified as one of the candidate targets of miR-199a-5p. To support this statement, recombinant plasmids of pGL3-SOCS7-3′ UTR-WT or pGL3-SOCS7-3′ UTR-Mut were constructed by inserting SOCS7-3′ UTR cDNA or its mutant (SOCS7-3′ UTR-Mut) into pGL3 and were cotransfected with miR-199a-5p (Fig. 4A). The luciferase reporter assay demonstrated that miR-199a-5p notably suppressed the luciferase expression of SOCS7-3′ UTR-LUC, while the mutation of 5 nucleotides in the 3′ UTR of SOCS7 resulted in the complete abrogation of the suppressive effect (Figs. 4B and 4C). Moreover, the SOCS7 mRNA level was notably repressed in LX2 cells after miR-199a-5p mimic treatment (Fig. 4D). Western blot analysis further demonstrated that SOCS7 protein expression was inhibited by miR-199a-5p (Figs. 4E and 4F). Functionally, SOCS7 inhibition resulted in an increased expression of collagen I and α-SMA in TGF-β1-activated LX2 cells (Supporting Figs. S1B and S1C).

Figure 4 SOCS7 was a direct target of miR-199a-5p.

(A) Schematic representation of the miR-199a-5p site in SOCS7-3′ UTR. (B) The expression of miR-199a-5p was assessed using qPCR analysis in LX2 cells transfected with miRcont inhibitor or miR-199a-5p mimic. (C) Luciferase activity was assayed in LX2 cells cotransfected with miR-199a-5p and luciferase reporters containing SOCS7-3′ UTR. Data are presented as the relative ratio of firefly luciferase activity to Renilla luciferase activity. (D) The mRNA expression of SOCS7 was assessed using qPCR analysis in LX2 cells transfected with NC or miR-199a-5p mimic. (E and F) Representative western blot of SOCS7 protein expression in LX2 cells transfected with miRcont or miR-199a-5p mimic, and quantification of average across three separate experiments. * P < 0.05, *** P < 0.001.

DZNep repressed fibrosis markers expression by upregulating SOCS7

Given that DZNep was able to inhibit miR-199a-5p expression in TGF-β1-activated LX2 cells and that SOCS7 was a direct target gene of miR-199a-5p, we investigated whether DZNep suppressed fibrosis markers expression by the mediation of SOCS7. To this end, we first assessed whether DZNep regulated SOCS7 expression in TGF-β1-activated LX2 cells. Figures 5A–5C showed that the SOCS7 mRNA and protein levels were decreased in LX2 cells after TGF-β1 treatment, whereas DZNep treatment partially inhibited TGF-β1-induced downregulation of SOCS7. SOCS7 was inhibited using SOCS7-specific siRNA (siSOCS7) in TGF-β1-activated LX2 cells, and then the expression levels of collagen I and α-SMA were measured in the presence or absence of DZNep. The expression of SOCS7 was significantly upregulated after DZNep treatment, whereas the effects were attenuated by the treatment with SOCS7-siRNA (Figs. 5D, 5G and 5H). More importantly, qPCR and Western blot analysis showed that the mRNA and protein expression of collagen I and α-SMA were decreased in the DZNep-treated group, whereas SOCS7 knockdown significantly decreased the role of DZNep in regulating collagen I and α-SMA mRNA expression (Figs. 5E–5J). Taken together, these data demonstrated that DZNep could effectively suppress hepatic fibrosis through regulating miR-199a-5p/SOCS7 pathway.

Figure 5 DZNep repressed liver fibrosis by upregulating SOCS7.

(A) The mRNA expression of SOCS7 was detected by qPCR analysis in TGF-β1-activated LX2 cells in the presence or absence of DZNep treatment. (B and C) Western blot and quantitative analysis of SOCS7 protein expression in TGF-β1-activated LX2 cells in the presence or absence of DZNep treatment. (D–F) The mRNA expression of SOCS7 (D), collagen I (E) and αSMA (F) was detected by qPCR analysis in TGF-β1-activated LX2 cells in the presence or absence of DZNep treatment after SOCS7 knockdown with siRNA-NC or SOCS7. (G–J) Representative Western blot of SOCS7 (G and H), collagen I (G and I) and α-SMA (G and J) in DZNep-treated LX2 cells in the presence of siRNA-NC (siNC) or siSOCS7, and quantification of average across three separate experiments. ∗P < 0.05, ∗∗P < 0.01.

Discussion

In this study, the effect of DZNep on repressing hepatic fibrosis and its underlying mechanism were further investigated. The current data verified that (i) DZNep alleviated CCl4-induced liver fibrosis in rats and in TGF-β1-activated HSCs; (ii) DZNep downregulated the expression of miR-199a-5p, and miR-199a-5p inhibition repressed TGF-β1-induced fibrosis markers expression; (iii) SOCS7 was a direct target of miR-199a-5p; and (iv) DZNep repressed fibrosis markers expression by upregulating SOCS7. These results uncovered the crucial role of the DZNep/miR-199a-5p axis in regulating liver fibrosis by regulating SOCS7 and may provide a therapeutic opportunity for patients with liver fibrosis.

Enhancer of zeste homolog 2 (EZH2) is a histone H3 lysine 27 (H3K27)-specific methyltransferase, and EZH2-mediated H3K27me3 is a suppressive posttranslational modification (McCabe et al., 2012). Zhang, Zhao & Zhu (2020) demonstrated that GAS5 recruits EZH2 to the matrix metalloproteinase 9 (MMP9) promoter region, thereby resulting in a subsequent downregulation of MMP9 expression, which relieves renal fibrosis in diabetic nephropathy rats. EZH2-mediated H3K27 hypermethylation at the cyclooxygenase-2 (COX-2) promoter results in the epigenetic silencing of COX-2, facilitating the fibrotic process in idiopathic pulmonary fibrosis (IPF) (Coward et al., 2014; Mozzetta et al., 2014). The effect of EZH2 on liver fibrosis has also been demonstrated. In murine or cell models of liver fibrosis, EZH2 expression is increased, and upregulated EZH2 inhibits Dickkopf-1 (an antagonist of Wnt/ β-catenin signaling) to activate Wnt/ β-catenin signaling and facilitate liver fibrosis (Miao et al., 2013; Yang et al., 2017). Therefore, the inhibition of EZH2 presents an excellent opportunity for regressing liver fibrosis. Martin-Mateos et al. (2019). demonstrated that pharmacological (GSK-503) or genetic (small interfering RNA) inhibition of EZH2 results in a significant decrease in TGF-β1-induced liver fibrosis markers. Carnosol, a compound extracted from rosemary, also shows an antifibrotic effect by repressing EZH2 (Zhao et al., 2018). Several small molecular inhibitors of EZH2, such as EPZ-6438, GSK126, and DZNep, have exhibited therapeutic potential in human diseases, including tumor and tissue fibrosis (Zeybel et al., 2017). In particular, the role of DZNep in inhibiting hepatic fibrosis progression has been verified. Zeybel et al. (2017) demonstrated that DZNep treatment is effective in repressing hepatic fibrosis progression by selectively targeting HSC-derived myofibroblasts. In this study, we verified that DZNep treatment effectively alleviated hepatic fibrosis progression in vitro and in vivo, but we further explored the specific mechanism by which DZNep alleviates liver fibrosis. Our data uncovered the specific mechanism of DZNep in the repression of fibrosis markers expression by regulating miR-199a-5p/SOCS7 axis.

Previous studies identified that increasing miR-200a level could alleviate liver fibrosis. Hu et al. demonstrated that IL-22 inhibits HSC activation and ameliorates liver fibrosis through enhancing expression of miR-200a and reducing expression of β-catenin (Hu et al., 2016). Sun et al. (2014) showed that miR-200a has an alleviative effect on HSC activation by decreasing the expression of β-catenin and TGF-β2. However, the results of this study indicated that the expression of miR-200a did not change significantly by DZNep treatment. Our data indicated that DZNep treatment inhibited the expression of miR-199a-5p, and downregulated miR-199a-5p contributed to repress TGF-β1-induced expression of fibrosis markers. The pulmonary miR-199a-5p level is upregulated in IPF patients and regulates TGF-β-induced lung fibroblast activation (Lino Cardenas et al., 2013). However, several studies have shown that exosomal miR-199a-5p suppresses the fibrogenic response by inhibiting connective tissue growth factor (CCN2) in HSCs (Chen et al., 2016). These studies discovered that the miR-199a-5p level is higher in quiescent HSC-derived exosomes than in activated HSC-derived exosomes, and exosomal miR-199a-5p is transferred from quiescent HSCs to activated HSCs and thus represses HSCs fibrosis. In this study, the miR-199a-5p level was markedly increased in LX2 cells after TGF-β1 treatment, whereas DZNep treatment repressed the increase. We next explored the role of miR-199a-5p inhibition in fibrosis markers expression, and our results indicated that downregulation of miR-199a-5p reversed TGF-β1-induced upregulation of collagen I and α-SMA. SOCS7 was identified as a novel target gene of miR-199a-5p. SOCS7 expression was decreased in TGF-β1-activated HSCs, but DZNep treatment restored SOCS7 expression. Importantly, SOCS7 inhibition weakened the effect of DZNep on collagen I and α-SMA expression in TGF-β1-activated HSCs.

Conclusions

The current study demonstrate that DZNep presents an antifibrotic effect by regulating the miR-199a-5p/SOCS7 pathway, and further study is essential to investigating whether DZNep can be applied as a therapeutic target for fibrosis.

Supplemental Information

Supplemental Information 1 Raw data

Click here for additional data file.

Supplemental Information 2 ARRIVE 2.0 Checklist

Click here for additional data file.

Figure S1 Supplemental Figure

Click here for additional data file.

List of abbreviations

DZNep 3-deazaneplanocin A

HSCs hepatic stellate cells

EZH2 enhancer of zeste homolog 2

SOCS7 suppressor of cytokine signaling 7

α-SMA Alpha-smooth muscle actin

Collagen I type I collagen

TGF-β1 transforming growth factor-β1

CCl4 carbon tetrachloride

ECM Extracellular matrix

H&E Hematoxylin and Eosin

DKK-1 dickkopf WNT signaling pathway inhibitor 1

NF- κB nuclear factor kappa B

ATCC American Type Culture Collection

FBS fetal bovine serum

DMEM Dulbecco Modified Eagle Medium

NC negative controls

SDS Sodium Dodecyl Sulfonate

ECL chemiluminescence

PFA paraformaldehyde

DAB 3,3′-diaminobenzidine

PBS phosphate buffer saline

SD standard deviation

ANOVA one-way analysis of variance

qPCR quantitative real-time Polymerase Chain Reaction

H3K27 histone H3 lysine 27

MMP9 matrix metalloproteinase 9

COX-2 cyclooxygenase-2

Additional Information and Declarations

Competing Interests

Author Contributions

Animal Ethics

Data Availability

The authors declare there are no competing interests.

Rongrong Ding, Jianming Zheng, Ning Li, Qi Cheng, Mengqi Zhu, Yanbing Wang, Xinlan Zhou, Zhanqing Zhang and Guangfeng Shi conceived and designed the experiments, performed the experiments, analyzed the data, prepared figures and/or tables, authored or reviewed drafts of the paper, and approved the final draft.

The following information was supplied relating to ethical approvals (i.e., approving body and any reference numbers):

The Animal Ethics Committee of Huashan Hospital, Fudan University approved all experiments in this study (2018 Huashan Hospital JS-030).

The following information was supplied regarding data availability:

The raw measurements are available in the Supplemental File.

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
