# Peer review of "DZNep, an inhibitor of the histone methyltransferase EZH2, suppresses hepatic fibrosis through regulating miR-199a-5p/SOCS7 pathway"

_PeerJ, doi:10.7717/peerj.11374_

## Round 0.1 · original submission · Major Revisions

Your study falls in a very interesting field and you will see that both reviewers have generally appreciated the idea of exploring EZH2 inhibitors as novel potential tools to treat hepatic fibrosis.

Both reviewers, however, find that several aspects of the work should be improved, i.e. in structuring the introduction, in the description of some of the methodology, in presenting the results, and also in some parts of the discussion that contain some overinterpretation.

In the light of these remarks, if you decide to submit a revised version, you must address all comments particularly from reviewer 2, who is providing very accurate guidance. Please pay particular attention to the following:

- Introduction: you may want to improve the logics in presenting your arguments;

- Figures: weaknesses in each figure are indicated and you must comply with suggestions from both reviewers;

- In terms of quality of the results, you need to clarify a major issue about statistics, i.e. either define normality if you use parametric tests or else use non-parametric tests if the distribution of your data is not normal;

- conclusions are overstated in places, and reviewer 2 is providing very sound arguments to explain where and why; please take notice;

- English language: major grammar mistakes are indicated, but please note that the list is not limited to the mistakes indicated by the reviewer. Please be sure that the revised final text reads in excellent scientific English.

Reviewer 1 ·

Basic reporting

Hepatic fibrosis is reversible damage to liver tissue and leads to the formation of Collagen I extracellular matrix. The study of molecular mechanisms by therapeutic drugs is novel ideas to understand the process of disease progression and improvement. Authors in this manuscript understand the molecular mechanism of a drug molecule DZNep in the prevention of hepatic fibrosis and propose to prevent liver cell injury and repair hepatic cells from damage in detailed observation. This study would help to understand the model of the DZNep axis in liver damage prevention and improve hepatic fibrosis by DZNep/miR-199a-5p/SOCS7 pathway. In addition, the authors propose that DZNep could act as a therapeutic agent to treat patients with reversible liver damage.

Experimental design

The authors applied qualitative and quantitative experimental designs to show the indispensable features of the DZNep axis. Further, they also applied in vitro and in vivo models to show the molecular basis of DZNep inhibition of liver damage. Moreover, the results in this manuscript are well presented according to their data. Quantitative analysis of results by authors displayed that DZNep is critical in regulating miR-199a-5p levels and in turn regulating SOCS7 levels.

Validity of the findings

Further, they presented several lines of evidence to show the DZNep effect on the SOCS7 pathway, like TGF-B1 levels, SMA levels, etc,. Based on the analysis I would recommend this study for PeerJ with the following improvements in the manuscript.

Additional comments

Major comments:
1. Figure 3A methodology and analysis are not clearly written. I suggest the authors to update methods for this section. Please also indicate the labels 1,2,3 in figure 3A.
2. Introduction can be improved by highlighting the role of SOCS7 in inflammation and how do SOCS7 levels abrogate liver fibrosis.
3. Authors may mention miR-199a-5p have complementary base pairing to SOCS7 3`UTR and they also indicate Figure 5A show mutated 3` UTR of SOCS7.
4. It would be very interesting if authors were able to show morphological differences DZNep treated or untreated HSCs or the effect of miR-199a-5p or SOCS7 levels on liver fibrosis phenotypes.
5. In the discussion maybe authors should propose a few best future directions to pursue in the DZNep field of liver fibrosis instead of claiming limitation of their study
Minor comments:
1. Add “Forward” for SOCS7 at line 128 in methods.
2. Line 219 miR-199a-5p should be “SOCS7”
3. Line 244 authors should mention where does H3K37me3 occurs in COX-2 gene, here statements seem to be incomplete.
4. Line 262 “Previous studies identified” is duplicated, please remove
5. In some figure legends (4E and F), a number of replicates were not mentioned, please indicate them in the next update.
6. In figures label DZNep (capitalization) appropriately to match the text in manuscript.
7. Please correct errors in the list of abbreviations:
FBS- fetal bovine serum
K-lysine not X
MMP9 repeated twice,

Reviewer 2 ·

Basic reporting

1) The English language should be improved in order to increase the readability of the manuscript. Ideally a scientist with high English proficiency should proofread the manuscript. Examples of sentences that require attention include but are not limited to
- Line 56: liver fibrosis can be regressed
- Line 58: Recent studies verified the effect
- Line 104: There were no survived rats
- Line 111: control (siNC) were structured by Thermo Fischer
- Line 243-244: EZH2 catalyzes H3K27me3 in cyclooxygenase-2
- Line 269: The pulmonary miR-199-5p level is in idiopathic form patients and regulates
- Line 278: miR-199-5p inhibition, observably inhibited
Throughout the manuscript “additional DZNep treatment” (as well as “additional treatment with SOCS7-siRNA” and similar statements) should be replaced by “DZNep treatment”.

2) The introduction shows the context and contains relevant references. However, this part could be better structured. Lines 72-73 state the aim of the study, then lines 74-79 present general information on miRNAs and lines 80-85 summarize the findings of the study. They authors may want to increase quality of their manuscript by improving their narrative.

3) Structure appears to conform to PeerJ standards.

4) While overall the Figures are appropriate and informative the labelling of Figures and data description should be improved.
- E.g. in Figure 1 (nor in the other figures) “NC” is not defined in the figure legends. It may be clearer to use “Control”. “Model” is not informative. Could be replaced by “CCl4” and “DZNep” could be replaced by “CCl4+DZNep” to enable readers to immediately understand the data.
- Figure 1: it is difficult to appreciate the subcellular location of the immunostaining in Figure 1: cytoplasmic or nuclear? What do the arrowheads point at? This data should be more clearly described in the results section and discussed. The most abundant cells in the liver are hepatocytes and likely the cells with positive EZH2 staining in the shown images are hepatocytes. However, the rest of the study is based on LX2 cells (fibroblasts) since fibroblasts play a major role in fibrosis. This is confusing and needs clarification. Additional immunostaining in cell culture experiments could be performed to assess the subcellular localization of EZH2 in LX2 cells in vitro.
- Molecular weights of proteins detected by Western blot should be indicated next to the images.
- Figure 3: the legends states “mRNA” expression for miRNA but per definition miRNA are not mRNA… this should be corrected in the manuscript wherever appropriate.
- Figure 5G should be replaced by a higher quality Western blot as it is one “representative experiments picked from” several technical replicates.
- Figure 5 is not easy to understand as in contrast to the other figures the effect of TGF-b1 is not represented as compared to control but TGF-b1 is represented as the control. Maybe the authors can find a way to both show that TGF-b1 increases in vitro markers of fibrosis and the effect of siSOCS2 in the presence and absence of DZNep. In Figure 2 the decrease in collagen I expression observed in the presence of DZNep is approximately 50% while in Figure 5 it appears to be only approximately 25%. The “rescue” of collagen I expression by siSOCS7 does not appear very robust, although an asterisk indicates statistical significance (please also see the comment on statistical analysis). Can this effect be reproduced when the initial decrease in collagen I is stronger?

Experimental design

1) The research question has been defined but could be presented in a more efficient way (see comment on the introduction).

2) The study design and statistical analysis need attention and appear not to meet the expected standards.
- The “Statistical analysis section” states that results are presented as means+/-SD. It should also include how many times each experiment has been performed. The figure legends indicate that representative images were picked from 3 separate experiments (technical replicates). Technical replicates are not independent experiments (biological replicates). Experiments are expected to have been performed e.g. 3 times on independent dates/with different cells using technical replicates within each of the experiments. Pooled data should show the data from the different experiments, not only technical replicates from one experiment.
- Furthermore, it states that statistical analysis was done using Student’s t test or ANOVA but there is no reference to the Normality test that should have been performed prior to the use of parametric tests. In case of a non normal distribution of data (which is commonly the case for small number of samples/data), non parametric tests should be used instead.

3) The methods are described in a global way and lack details in order to enable reproduction of data. Furthermore, the rational for some experiments should be indicated.
- E.g. lines 111-115 the amount of miRNA mimics/inhibitors and the timing of transfection relative of cell plating and readout of the experiment should be clearly stated.
- Lines 131-and following: the amount of protein used for Western blot experiments should be clearly indicated.
- Lines 139-143: how much vector was transfected? At what time following plating of the cells? For dual luciferase reporter assay LX2 cells were used, i.e. cells that express miR-199-5p. Why did the authors choose this cell line to perform the reported assay? Usually, a cell line that does not express the miRNA of interest (e.g. as assessed by PCR) is chosen in order to avoid background inhibition of the reporter by the endogenous miRNA.

Validity of the findings

1) As indicated above, experimental design/data analysis/representation need attention.
- While appropriate controls (control siRNA/miRNA/inhibitors) appear to have been used in most of the experiments, the data cannot be seen as robust and statistically sound at this stage (also see comments on statistical analysis).
- In Figure 2, the authors did not include a control of DZNep in control cells. What is the effect of DZNep in cells not treated with TGF-b1?

2) Several conclusions appear to be overstated.
- They authors should more carefully phrase their conclusions in order to limit their conclusions to the shown results. E.g. an effect on collagen I and a-SMA shown in vitro with LX2 cells cannot be interpreted as an effect on fibrosis as fibrosis relates to pathological changes seen in an organ (as stated by the authors in their introduction). Therefore, throughout the manuscript the authors should replace “hepatic fibrosis” (or equivalent term) when referring to in vitro data by more appropriate description of their data.
- Since the authors have access to an animal model of hepatic fibrosis, why did they not also assess the role of miRNAs in vivo in rat livers? It should be clearly stated why they solely chose LX2 cells for their mechanistical study. Is miR-199-5p conserved between humans and rats? What is miR-199-5p/SOCS7 expression in rat liver following DZNEp treatment and control condition?
- Line 220 description of Fig 5A: “partially” should be added to read “DZEp treatment partially inhibited TGF-b1-induced downregulation of SOCS7”
- Furthermore, the authors base all their conclusions related to EZH2 on DZNep, an inhibitor that globally inhibits histone methylation that is known not to be selective for EZH2. Indeed, while it was the first EZH2 inhibitor that has been developed, several other more selective EZH2 inhibitors have been developed since 2012 and it would be more appropriate to use one of those EZH2 inhibitors, e.g. an inhibitor in clinical development. Given the absent selectivity of DZNep, it is difficult to conclude about a direct role of EZH2 in TGF-b1-induced increase of collagen I/a-SMA. These data should be confirmed using a more selective EZH2 inhibitor, siEZH2 and/or EZH2 rescue experiments.

Additional comments

This study deals with an interesting topic: the molecular mechanisms of hepatic fibrosis and the potential of EZH2 inhibitors for treatment of liver fibrosis. While this is a very interesting field of research, the present study suffers several flaws and could be improved by addressing the comments detailed above.

---

## Round 0.2 · Minor Revisions

While the reviewer has appreciated an improved presentation of the data and of the English language in the revision, you will find that the reviewer also noticed that some points have not been addressed.

- The statistical analysis of the Westerns is unclear: in the rebuttal you write “separate experiments (technical replicates)”. That is contradictory: either you have performed independent experiments, or else you have technical replicates of one same experiment. Please be consistent in your reply as you know that your original scans may be required.

- Please check the molecular weight markers in the Western blots in Figure 5B and Figure 5G.

- Please comment on the reviewer’s remark that It could have assessed miR-199-5p expression in rat livers with and without DZNEp treatment to extend the validity of the findings in cultured cells, while the rat model in Figure 1 per se does not provide new findings.

- The reviewer noticed that the decrease in collagen I expression observed in the presence of DZNep (Figure 2) is different from the effect shown in Figure 5. Your reply that “We did not know how to explain the phenomenon though the results from both experimental results are significantly different” is not adequate. If a phenomenon cannot be explained, then what is its added value?

- In addition, the newly added immunofluorescence data need a better description. Supporting Figure 1 also needs a legend.

- Also please pay particular attention to the last sentence (“indicate that DZNep may be applied as a therapeutic target for fibrosis”). The idea that DZNep could be used as a therapeutic strategy is not actually supported and this should be rephrased.

Reviewer 2 ·

Basic reporting

The English language and narrative have been improved thereby increasing the readability of the manuscript.

The title is an overstatement since the in vitro data have not been linked to the mouse model and could be toned down to better reflect the actual data.

The labelling of Figures and data description has been improved. Nevertheless, there are mistakes in the molecular weight description of the Western blots that should be corrected (in Figure 5B SOCS7 is labelled 63 kDa and in Figure 5G 85 kDa).

Experimental design

Statistical analysis section has been updated to include all tests that were used. It also states that means+/- SD were calculated on several independent experiments. Nevertheless, it is still not clear whether Western blot quantifications have been performed on independent experiments or solely on technical replicates from one experiment as the figure legends indicate that “representative images were picked from 3 separate experiments (technical replicates)”. This should be clarified.

The methods section has been updated and now includes more details in order to enable reproduction of the experiments. However, it does not include the description of immunofluorescence analysis that were added to Figure 1 nor the new data that are presented in the new Supporting Figure 1. This should be added.

The authors should include a legend for the new Supporting Figure 1. Furthermore ‘TGFb1” should be added to panels A and B of this Figure (as done in Figure 5) to clearly indicate that the experiment was done in the presence of TGFb).

Validity of the findings

Using DZNep in control cells would enable to assess the role of EZH2 in basal expression of the genes studied in addition to TGF-beta induced gene expression.

The authors indicate that miR-199-5p is conserved between humans and rats. It would thus have been easy to assess miR-199-5p expression in rat livers following DZNEp treatment and control condition to extend their finding in LX2 cells to the rat model shown in Figure 1 that does by itself not provide new findings as compared to previously published data and is not directly related to the rest of the figures.

Since the authors agree that it would be interesting to confirm their data obtained with DZNep using a more selective EZH2 inhibitor, siEZH2 and/or EZH2 rescue experiments, it would have been interesting to provide at least a few additional data.

Last sentence (“indicate that DZNep may be applied as a therapeutic target for fibrosis”) should be rephrased. According to the authors, DZNep would be used as a therapeutic strategy (no target…).

Additional comments

The English language, narrative, and data presentation have been improved in the revised manuscript.
Nevertheless, there is still some information lacking, particularly regarding the newly added data.
The manuscript could have been further strengthened by addressing the previous comments that the authors chose not to address.

---

## Round 0.3 · accepted · Accept

Thank you for re-revising your work in compliance with the reviewer's suggestion.